# Comparative Analysis of *PRNP* Gene Indel Polymorphism and Expression among Zhongdian Yellow Cattle, Zhongdian Yak, and Their Hybrids

**DOI:** 10.3390/ani13233627

**Published:** 2023-11-23

**Authors:** Xiaoming He, Sameeullah Memon, Dan Yue, Junhong Zhu, Ying Lu, Xingneng Liu, Heli Xiong, Guozhi Li, Weidong Deng, Dongmei Xi

**Affiliations:** 1Yunnan Provincial Key Laboratory of Animal Nutrition and Feed, Faculty of Animal Science and Technology, Yunnan Agricultural University, Kunming 650201, China; xiaominghe@foxmail.com (X.H.); danyue0528@foxmail.com (D.Y.); junhong-zhu@foxmail.com (J.Z.); yinglu_1998@163.com (Y.L.); helihewei@163.com (H.X.); guozhi_li2023@163.com (G.L.); dengwd@ynau.edu.cn (W.D.); 2Korea Zoonosis Research Institute, Jeonbuk National University, Hana-ro, Iksan 820-120, Jeonbuk, Republic of Korea

**Keywords:** *PRNP* gene, gene expression, BSE, Zhongdian Yak, Zhongdian Yellow cattle, Zhongdian Yakow

## Abstract

**Simple Summary:**

Bovine spongiform encephalopathy (BSE), a neurological disorder with a significant effect on cattle health, has been recognized pathologically and medically. It is associated with a 12 bp indel in intron 1 of the *PRNP* gene and a 23 bp indel polymorphism in the putative promoter. The main focus of this work was to assess the allele, genotype, and haplotype frequencies of *PRNP* indel polymorphisms in Zhongdian Yak, Zhongdian Yellow cattle, and Zhongdian Yakow, as well as to investigate the effect of *PRNP* gene expressions of 23 bp and 12 bp indels using PCR. Two variable sites—a 23 bp indel polymorphism holding AP1 binding site and a 12 bp indel polymorphism holding SP1 binding site—were also investigated. Overall, the study results suggest that the *PRNP* genotype may contribute to the high variation in *PRNP* expression.

**Abstract:**

Bovine spongiform encephalopathy (BSE) is a fatal disease in cattle caused by misfolded prion proteins and linked to indel polymorphisms in the promoter and intron 1 of the *PRNP* gene. The aim of this study was to determine the allele, genotype, and haplotype frequencies of *PRNP* indel polymorphisms and to investigate the effect of *PRNP* gene expressions of 23 bp and 12 bp indels via polymerase chain reaction (PCR) in Zhongdian Yak (*Bos-grunniens*) (YK), Zhongdian Yellow cattle (*Bos-taurus*) (YC), and Zhongdian Yakow (*Bos-primigenius taurus* × *Bos-grunniens*) (PK). Resultant high allelic frequencies were found in 23− and 12+, while haplotype frequencies were very low in 23+/12 in YK, YC, and PK. *PRNP* expression was higher in the +−/−− diplotype of the PK and (mean ± SE) was 3.6578 ± 1.85964. Furthermore, two variable sites were investigated—a 23 bp indel polymorphism holding AP1 binding site and a 12 bp indel polymorphism holding SP1 binding site. Additionally, reporter gene assays revealed a link between two proposed transcription factors and lower expression levels of the +/+ allele compared with the −/− allele. The expression level of *PRNP* was shown to be dependent on two indel polymorphisms in the bovine *PRNP* promoter, which includes binding sites for RP58 and SP1 transcription factors. These findings raised the possibility that the *PRNP* genotype may contribute to the high variation in *PRNP* expression.

## 1. Introduction

Prion diseases, also known as transmissible spongiform encephalopathy (TSE), are a category of mammalian neurodegenerative diseases that affect ruminants, humans, cats, and mink [1]. The prion protein gene (*PRNP*) encodes the host-encoded cellular prion protein (PrP^C^), which plays an important role in prion disease and prevents the development of the disease or the transmission of the disease when PrP^C^ is absent [2]. TSE susceptibility is lowered when *PRNP* expression is low or mild [3]; however, *PRNP* expression is high while susceptibility is increased and incubation time is reduced prior to disease development [4]. TSE is mostly defined by PrP^C^, which undergoes structural isomerization to produce an infectious form known as prion protein scrapie PrP^Sc^, which accumulates pathologically [5]. PrP^C^ has been found in a variety of tissues, and it is expected to play a key role in copper metabolism in the central nervous and immune systems [6].

BSE has been conspicuously observed in prion diseases due to various human infections causing variant vCJD [7]. Among these, two indel polymorphisms have been identified, which are characterized by a 23 bp indel in the putative promoter and a 12 bp indel in intron 1, as well as a number of octapeptide repeats (octarepeats) in the coding sequence and polymorphism in amino acids [8]. Normally, it is hypothesized that the frequencies of indel polymorphisms at two non-coding regions of bovine *PRNP*—23 bp indel and 12 bp indel sites—are associated with PrP^C^ levels [9,10]. The RP58 repressor protein binding site was eliminated by a 23 bp deletion within the higher region of the promoter in the first polymorphism, and the SP1 transcription factor binding site was eliminated by a 12 bp deletion within intron 1 [9]. A reporter gene assay has shown an interaction between the two postulated transcription factors and lower expression levels at 23 ins/12 ins alleles compared with the 23 del/12 del alleles [9]. Additionally, it has been noted that cattle with these deletions are more prone to classical BSE due to the lack of binding sites for the relevant regulatory elements [11], and these polymorphisms have no influence on resistance to atypical BSE [11,12,13]. Moreover, indel polymorphisms that affect classical BSE susceptibility may not seem to apply to other transmissible spongiform encephalopathies in cattle [14].

Polymorphism frequencies in the *PRNP* gene promoter region have been reported in cattle in Asia [10,15,16,17,18,19,20,21], Europe [11,12,22,23,24,25,26], and the Americas [27,28]. Additionally, new facts and figures regarding the distribution of bovine *PRNP* indel polymorphisms and frequency have been recorded for Vietnamese local cattle and native Chinese cattle [21], and the data show that Asian local breed cattle have many kinds of genetic divergences within the *PRNP* for potential association with BSE. In a study in UK Holstein cattle, it was reported that the 12 bp indel is the core recipe of resistance to classical BSE [11], while the mRNA levels of the 23 bp indel were found in higher numbers in the medulla oblongata of Japanese Black cattle [10].

The aims of the present study were to identify the allele, genotype, and haplotype frequencies of the *PRNP* indel polymorphisms and examine the effect of *PRNP* gene expressions of 23 bp and 12 bp indels via polymerase chain reaction in Zhongdian Yak (YK), Zhongdian Yellow cattle (YC), and Zhongdian Yakow (PK) from the Yunnan province of China.

## 2. Materials and Methods

### 2.1. Sample Collection and DNA Extraction

In this study, we used medulla oblongata tissue taken from 562 Zhongdian Yak (YK), 368 Yellow cattle (YC), and 377 Zhongdian Yakow (PK) animals that were slaughtered in Shangri-La City, China. The samples were taken randomly and summarized according to age and sex. The YK, YC, and PK samples were divided into three groups based on their ages: 1–6, 7–13, and 14–20 months. RNA preservation solution was added to tissue samples measuring 0.3 cm^2^ that were placed in 1.8 mL freezer storage tubes and kept at −20 °C in the refrigerator. An animal DNA extraction kit (Beijing full-type Gold Biotechnology Co., Ltd., Beijing, China) was used for DNA extraction following the manufacturer’s instructions.

### 2.2. RNA Isolation, RNA Purity, Concentration, Integrity Detection, and cDNA Synthesis

Total RNA was extracted from 500 mg of the freeze-dried medulla oblongata tissue samples using the total RNA extraction kit (Tian Gen Biological Technology Co., Ltd., Beijing, China) as per the manufacturer’s instructions. The 3 µL of RNA solution was dissolved in 297 µL of Rnase-free water, and the concentration and purity of RNA were detected by using Rnase-free water as a blank control and a nucleic acid protein analyzer. The purity of RNA was assessed via the absorbance of OD value (od^260/280^ and A^260/280^) ratios between 1.8 and 2.1, which suggested the RNA was free of contamination. RNA integrity was checked via agar gel electrophoresis using 1% agarose gel. The agarose gel analysis of DNA consisted of voltage (V) and electrophoresis reverse transcription synthesis of cDNA using the reverse record kit (Beijing full-type Gold Biotechnology Co., Ltd.).

### 2.3. Construction of a Standard Curve and Real-Time Reverse Transcription PCR Assay

The PCR used to amplify the standard curve was carried out using the GeneAmpH PCR System 9700 as shown in Appendix A. Gene expression levels of the bovine *PRNP* and ACTB were quantitatively measured via real-time PCR using the primer pairs listed in Appendix A. The real-time PCR technique was carried out in a 15 mL reaction volume containing 7.5 mL of SYBRH Green Real-Time PCR Master Mix (TOYOBO, Osaka, Japan), 0.03 mM of each primer, and 3.0 mL of each cDNA in the 7300 Real-Time PCR System (Applied Biosystems). Thermal cycling conditions were 95 °C for 10 min followed by 40 cycles of 95 °C for 10 s and 60 °C for one minute. All reactions (standard, unknown samples, and non-template control) were performed in duplicates using the same 96-well plate. The results reported here are averages of the duplicates.

### 2.4. Establishment of the Standard Curve of the ACTB Gene

After the concentration of the samples was measured, the sample copy number was calculated by the formula, and the recovery sample of 1 µL was divided into 7 gradients and diluted 110 times. The diluted sample was used as the template, and the standard curve of fluorescence quantitative PCR was constructed. It is well known that the ACTB gene fragment length is 143 bp and that the ACTB gene gum recovery concentration is 3300 ng/µL in the UV–visible nucleic acid protein analyzer. The number of relative templates for quantitative PCR products per microliter of ACTB fragment is determined as follows: copy number = concentration (ng/µL) × 1 µL × 6.02 × 1023/mol/[fragment length (bp) × 660 g/(MOLXBP)]. Because each base of the microsegregation was 330 and each base bp was 660 G/mol, the Avegadro constant is 6.02 × 1023/mol (the number of particles per mol). Therefore, each standard sample contained the number of templates in the following order: 2.10 × 10^21^, 2.10 × 10^20^, 2.10 × 10^19^, 2.10 × 10^18^, 2.10 × 10^17^, 2.10 × 10^16^, 2.10 × 10^15^, and so on. Subsequently, the standard curve of the ACTB gene was constructed via fluorescence quantitative PCR with each sample of different concentrations as a template.

### 2.5. Establishment of the Standard Curve of the PRNP Gene

The *PRNP* gene fragment length is known to be 128 bp, and a UV–visible nucleic acid protein analyzer was used to find the *PRNP* gene gum recovery concentration of 2 ng/µL. The gradient of the standard curve of the absolute template *PRNP* gene was the same as that of the relative ACTB gene: 1.99 × 10^21^, 1.99 × 10^20^, 1.99 × 10^19^, 1.99 × 10^18^, 1.99 × 10^17^, 1.99 × 10^16^, and 1.99 × 10^15^. Thus, the standard curve of the *PRNP* gene of the absolute template was constructed.

### 2.6. Promoter–Reporter Gene Constructs

All the gene constructs were arranged and are displayed in Appendix A. PCR fragments for pDel/Del, pDel/Ins, pIns/Del, and pIns/Ins including the 5_flanking sequence, exon1 (EX1), intron 1, and the first part of exon2 (EX2) of the bovine *PRNP* gene were produced by using a forward primer 5′-AGACGCGTTGCCCAGCCCAGGTGCCAGCCAT-3′ and a reverse primer 5′-GCAGATCTATCTGCTGTGATTCAGCTCAAGTT-3′. The constructs p12Ins and p12Del were shaped via elongated PCR amplification (Thermo Fisher, Santa Clara, CA, USA) according to the manufacturer’s protocol with a touch-down protocol using a forward primer 5′-ATCTCGAGGGGTCTGCCAGTAAACCCCGGGCG-3′ and a reverse primer 5′-GCAGATCTATCTGCTGTGATTCAGCTCAAGTT-3′; the lowercase letters specify the restraint sequence of Mlu-I and Bgl-II, correspondingly. PCR products were duplicated into a pGL3-Basic Vector luciferase reporter vector (Promega) by using restriction endonucleases Mlu-I and Bgl-II for pDel/Del, pDel/Ins, pIns/Del, and pIns/Ins, or Xol-I and Bgl-II for p12Ins and p12Del. The constructs were sequenced using an ABI Prism-310 Genetic Analyzer (Applied Biosystems Inc., Foster City, CA, USA).

### 2.7. Cell Culture and Transfection

Neuroblastoma cells (N2a obtained from the American Type Culture Collection (ATCCH), Number: CCL-131TM) were cultured in Eagle’s minimum essential medium with non-essential amino acids and sodium pyruvate and supplemented with 10% fetal calf serum at 37 °C under 5% CO_2_ for the luciferase assay. In order to perform the luciferase assay, N2a cells were seeded at 6 × 10^4^ cells/well in 24-well plates 48 h before transfection. Cells reaching 60–80% confluence were co-transfected with 540 ng of bovine *PRNP* promoter luciferase vectors and a pRL-TK vector, or an empty pGL3/pRL-TK vector. The transfections were carried out using Lipofectamine^TM^ LTX and PLUS^TM^ reagents according to the manufacturer’s protocol (Invitrogen, Carlsbad, CA, USA).

### 2.8. Luciferase Assay

The luciferase activity of cell lysates organized at 48 h after transfection was calculated as virtual light units with the TriStar LB 941 Multimode Reader (Berthold Technologies, Bioanalytic, Bad Wildbad, Germany) using the Dual-Luciferase Assay System (Promega, Madison, WI, USA). Relative luciferase activities were explained as the ratio of firefly luciferase activity to that of the Renilla luciferase, which holds the HSV-TK promoter. The association of the firefly luciferase activity and the bovine *PRNP* promoter was evaluated by comparing the constructs with the empty pGL3 vector.

### 2.9. Electrophoretic Mobility Shift Assay (EMSA)

Nuclear extracts were arranged from bovine brain and PT cells transfected with the vector pRP58 following the protocol by Dignam et al. [29]. Approximately 50 bp oligonucleotides surrounding polymorphisms_1980T3C, _1594indel23bp, _85G3T, _300indel12bp, _571A3G, and _709A3G were designed with both alleles. The oligonucleotides with sequences near the polymorphism_571A3G were used for EMSA with a gel shift assay system (Promega, Mannheim, Germany) according to the manufacturer’s directions. The remaining oligonucleotides were double stranded, labeled with 32 P, and applied at 25 fmol for DNA-protein binding reactions with 1 g of poly, 10 g of bovine serum albumin, binding buffer (250 mM HEPES/NaOH (pH 7.9), 50 mM MgCl_2_, 750 mM to 1.5 M NaCl, 5 mM dithiothreitol, 5 mM EDTA, and 25% glycerol), 2.25–8.75 g of nuclear extract or 300 ng of recombinant human SP1 extract (Promega), and either 250 fmol or 2.5 pmol of unlabeled double-stranded oligonucleotides as specific or non-specific competitors. The binding reactions were incubated for 10 min on ice. Electrophoresis of the samples through a native 8% polyacrylamide gel (19:1 acrylamide/bisacrylamide) in 1% Tris borate/EDTA buffer was followed by autoradiography. The sequences used for the EMSA probe are shown in Appendix A.

### 2.10. Statistical Analysis

The calculation of genotype frequency and gene frequency used N two-fold biota, with one gene set containing two alleles (e.g., aa); these two alleles were composed of the genotype of AA, AA and AA, whereas AA was a dominant homozygous individual with D, AA was a mixed individual with H, and AA was a recessive pure individual with R. Therefore, d + h + r = *n*, the allele number was 2 N, and the allele frequency and gene frequency were calculated. Data processing and analysis of the *PRNP* gene via RT-PCR data were analyzed using variance analysis, a significance test, and correlation analysis using SPSS20, and the results were expressed as the average ± standard error. The program Haploview 4.0 was applied to analyze the haplotype frequencies derived from the genotypic data. Excel 2010 software was used to process data and chart production. Differences between constructs were tested for significance using the procedure MIXED of the SAS 9.4 program (SAS Institute Inc., Cary, NC, USA).

## 3. Results

### 3.1. Genotypic and Allelic Frequencies

We genotyped a total of 1307 animals including 662 YK, 368 YC, and 377 PK to investigate *PRNP* gene polymorphisms in the 23 bp and 12 bp indel loci. Both 23 bp and 12 bp loci were polymorphic in three species. It was observed that alleles of 23 bp deletion in YK, YC, and PK were high in −/− frequencies (0.838, 0.399, and 0.467), whereas the alleles of 12 bp insertion were high in +/+ frequencies (0.767, 0.621, and 0.714), respectively. Genotypic and allelic data are shown in Table 1.

### 3.2. Haplotypic Frequencies

In this study, the four different haplotypes (23+/12+, 23+/12−, 23−/12+, and 23−/12−) were found in the *PRNP* gene promoter and intron 1 of three different species, YK, YC, and PK (Table 2). It was observed that haplotype 23−/12+ was in high frequency compared with the other three haplotypes in three different species. The most frequent haplotype obtained was 23−/12+ with a high frequency of 0.796 while the haplotype 23+/12+ showed the lowest frequency in YK compared with the two other species (Table 2).

### 3.3. Effect of PRNP Expression Based on Sex and Age of the 23 bp Promoter Indel and 12 bp Intron 1 Region

Expression of the *PRNP* gene compared with sex and age in YK, YC, and PK was evaluated. Our result for the male samples (mean ± SE) was 1.6936 ± 0.23701, and the result for the female samples (mean ± SE) was 1.6664 ± 0.22736. We observed greater *PRNP* expressions of PK compared with the other two species of YK and YC, whereas a low expression of *PRNP* was detected both in males (mean ± SE) at 1.3893 ± 0.08583 and in females (mean ± SE) at 1.2053 ± 0.11304 of YK with respect to sex (Figure 1). Furthermore, the comparison of *PRNP* expressions showed age differences of 1–6 months, 7–13 months, 14–20 months, and a total of 1–20 months among the three species of PK, YC, and PK. As a result, low *PRNP* expressions were detected in PK and YK compared with YC, whereas the (mean ± SE) was found to be 2.2118 ± 0.26173, which is the highest *PRNP* expression in PK compared with other species (Figure 2).

### 3.4. Effect of PRNP Expression Based on Genotype of the 23 bp Promoter Indel

To estimate the expression of *PRNP* among genotypes of the 23 bp promoter indel, three genotypes were examined from YK, YC, and PK. The expression of the *PRNP* gene relative to the indication ACTB gene was analyzed, and the genotypes investigated were ++, +−, and −− of the 23 bp locus. *PRNP* expression was found to be higher in the −− genotype of the YK and (mean ± SE) was 0.8100 ± 0.04319, whereas the higher *PRNP* expression was also found in the −− genotype of the PK and (mean ± SE) was 4.1952 ± 1.23670; similar higher *PRNP* expressions were found in the − diplotype of the YC and (mean ± SE) was 3.6042 ± 1.06625, respectively (Table 3).

### 3.5. Effect of PRNP Expression Based on Genotype of the 12 bp Indel in Intron 1

To estimate the expression of *PRNP* among genotypes of the 12 bp indel intron 1, three genotypes were examined from YK, YC, and PK. The expression of the *PRNP* gene relative to the indication ACTB gene was analyzed, and the genotypes investigated were ++, +−, and −− of the 12 bp locus. *PRNP* expression was higher in the − genotype of the YK and (mean ± SE) was 1.1373 ± 0.24600, while higher *PRNP* expression was also found in the +− genotype of the PK and (mean ± SE) was 6.1033 ± 2.89163; the same higher *PRNP* expression was found in the − diplotype of the YC and (mean ± SE) was 3.2614 ± 0.79525, respectively, as shown (Table 4).

### 3.6. Effect of PRNP Expression Based on Genotypes of the 23 bp Promoter Indel and 12 bp Intron 1 Region

To evaluate the expression of *PRNP* among genotypes of the 23 bp promoter and the 12 bp intron 1 indel, six diplotypes were investigated from PK and YK, and eight genotypes were investigated from YC. The diplotypes comprised genotypes ++, +−, and −− of the 23 bp locus on the left side, whereas the 12 bp locus is a parallel genotype in all diplotypes on the right side. It was assumed that a parallel genotype at the 12 bp locus would have a similar cause in these diplotypes and, for that reason, the gene expression was compared along with genotypes of the 23 bp locus. The expression of the *PRNP* gene relative to the indication ACTB gene was analyzed. The *PRNP* expression was higher in the −−/−− diplotype of the YK and (mean ± SE) was 2.8820 ± 0.62277, whereas a higher *PRNP* expression was found in the +−/−− diplotype of the PK and (mean ± SE) was 3.6578 ± 1.85964; a similar higher *PRNP* expression was found in the +−/+− diplotype of the YC and (mean ± SE) was 3.4060 ± 1.21839, respectively (Table 5). Therefore, in comparison with the total sample of each breed, the expression of *PRNP* was observed to be the highest in the +−/−− diplotype and (mean ± SE) was 3.6578 ± 1.85964 of the PK—higher than in the YK and YC—while compared with the other two species of PK and YK, the lowest *PRNP* expression was found in the −−/−− diplotype and (mean ± SE) was 1.0111 of the YC (Figure 3).

### 3.7. Binding of Putative Transcription Factors to Selected Sections of the PRNP Promoter

We examined the influence of 12 bp and 23 bp indels and SNP on the binding activity of the transcription factor SP1 to the *PRNP* gene promoter. Gel shift assays were carried out with nuclear extract from N2a cells. The remarkable variations in transcription factor binding to dissimilar *PRNP* alleles were examined in the case of 12 bp and 23 bp indel polymorphisms and the transcription factor AP1. The 23−/12−, 23−/12+, 23+/12+, and 23+/12– haplotypes were found to bind AP1, whereas the other four did not bind AP1 (Figure 4). EMSA was carried out with prospective SP1 binding sites within 23 bp and 12 bp indels and SNP on the binding activity of the transcription factor SP1 to the *PRNP* promoter, and gel shift assays were carried out with nuclear extract from N2a cells. Additionally, the same haplotypes, 23−/12−, 23−/12+, 23+/12+, and 23+/12–, were found to bind SP1 (Figure 4).

### 3.8. Putative Transcription Factor Binding Sites and Promoter–Reporter Gene Assays

The bovine *PRNP* promoter region position 4270 was displayed in silico for transcription factor binding sites, and it was revealed that both sites contributed to promoter activity [30]; this region comprises a 5_flanking sequence also called intron 1 (Appendix A). Plasmids pDelDel, pDelIns, pInsDel, pInsIns, p12Ins, and p12Del conducted the firefly luciferase open reading frame. Under the control of dissimilar *PRNP,* promoter alleles were co-transfected into two dissimilar bovine cell lines, PT and KOP, with the pRL-TK normalization vector (Figure 5). For every polymorphic location, each allele was displayed for transcription factor binding sites, leading to the detection of 44 divergences in putative transcription factor binding sites. Discrepancy sites based on possessions of potentially binding transcription factors were expelled. For instance, a discrepancy transcription factor binding site of hepatic nuclear factor-4 was not pursued. This transcription factor is occupied generally in liver-specific gene regulation. Ultimately, three transcription factor binding sites determined to be functionally important for the regulation of *PRNP* transcriptions were selected (Appendix A).

## 4. Discussion

BSE is a fatal neurodegenerative disorder caused by abnormal metabolism of prion protein (PrP) [30,31,32,33,34,35]. It has been reported that these cellular prion proteins (PrP^C^) are encoded by prion proteins whose expressions show a vital role in prion disease susceptibility [36]. Moreover, the 23 bp (23−) and 12 bp (12−) deletions have been previously observed to be linked with *PRNP* expression [9,10] and BSE susceptibility [11,12,13]. Furthermore, 12 bp indel polymorphisms in intron 1 and 23 bp in promoter regions are components of mutations that occur in the *PRNP* gene, resulting in disease susceptibility [26]. It has been observed that the −−/−− diplotype has been reported in high frequency in BSE-affected animals [8,37]. The present study has also demonstrated that the 12 bp indel polymorphisms in intron 1 and 23 bp in promoter regions are linked with susceptibility of BSE in Non-*Bos taurus* animals within the *Bos* genus.

The 23 bp and 12 bp deletion alleles were linked with a high vulnerability to BSE as reported in previous studies [11,12,38]. The results of the present study demonstrate that genetic distributions for YK, YC, and PK have shown parallel models crosswise with healthy populations in the regions. Previous studies have shown that the 23 bp and 12 bp deletion alleles, the 23−/− and 12−/− genotypes, and the 12−/23− haplotype were linked with high vulnerability of European cattle to classical BSE [11,12,13,36,38]. In contrast, the occurrence of insertion alleles at the 23 bp and 12 bp indel loci is linked to classical BSE resistance. Additionally, the 12+/23+ haplotypes were also linked with BSE resistance, and the resistance was measured to be at the 12 bp indel locus [11,22]. Consequently, haplotype analysis observed that the 12 bp indel is the most significant cause of BSE vulnerability. However, our results may not conclusively prove that each or both indel polymorphisms play a similar role in the etiology of BSE. The evidence of intron 1, where the 12 bp indel locus was located, shows that it plays a role in controlling *PRNP* gene expression levels, which was examined in previous studies on *PRNP* gene regulation [10,22,37,38].

Diplotypes were used to examine the effect of genotypes of the 23 bp and 12 bp loci on the *PRNP* gene expression in this study. Initially, we revealed that diplotypes with +−/+− and +−/−− genotypes were found in high expression compared with those with ++/++, ++/+−, ++/−−, and −−/−− genotypes among the three breeds. Moreover, we calculated the divergence of *PRNP* gene expression among diplotypes varying in one allele (+ or −) at the 23 bp and 12 bp loci, and the results demonstrated that diplotypes with the + allele resulted in a lower *PRNP* gene level compared with the − allele. The +− genotype showed a higher *PRNP* gene level than the other two genotypes because of possible overdominance, whereas the − allele might be covered with the + allele due to other factors, including polar overdominance as found in other breeds [39]. Previous studies have reported that there is incomplete information on the overdominance of the + allele at the 12 bp locus in the cattle *PRNP* gene. However, in comparison with the other two genotypes, the +− genotype has been previously observed in high frequency at the 12 bp locus in several cattle populations [8,10]. In comparison with the homozygous genotypes ++/++ and the −−/−− diplotypes, our findings indicate that the +−/+− and +−/−− genotypes of the 23 bp and 12 bp loci may contribute a higher *PRNP* gene level. Since these loci were strongly located in a region of higher linkage disequilibrium in bovine *PRNP*, these were not easily distinguishable [40]. Hence, the condition is comparable to the expression of prion protein, and the levels of *PRNP* may illuminate the levels of prion protein in the medulla oblongata. This study demonstrated results similar to a previous report that showed that divergence between the + and − alleles at a similar location in two diplotypes might drastically influence the level of the *PRNP* [10]. Our findings revealed that the increase in the − allele increased the level of *PRNP* in one of two diplotypes observed. It is critical to note that the ++/++ diplotype was found to have a low *PRNP* level compared with other diplotypes in all three breeds, which suggests that a ++ of both loci might show a lower *PRNP* expression.

Furthermore, the expression of the *PRNP* gene in the medulla oblongata was estimated with the impact of sex and age from YK, YC, and PK. The expression of *PRNP* was found to be high in the male samples of YK and PK, while low *PRNP* expression was found in the YC male samples compared with the female samples, with no significant difference between them. In terms of age differences, high *PRNP* expression was found in PK with respect to all age differences compared with YK and YC. The findings showed a similar pattern to that described by [10], suggesting that the expression of the *PRNP* gene was generally identical and unaffected by age, sex, or coat color.

This study was conducted in a region that included a *PRNP* promoter, which has been investigated in previous mutation analyses of the bovine *PRNP* gene [9,12,41]. Our research was theorized with the purpose of investigating if changes influencing the *PRNP* gene expression level might have an effect on BSE vulnerability. The 12 bp indel polymorphisms in intron 1 and the 23 bp indel polymorphisms in the 5_flanking region were shown to be uncertainly related to BSE vulnerability in *PRNP* genotypes. The 12 bp indel polymorphism and certain single nucleotide polymorphisms within the putative promoter region are in sturdy association disequilibrium with the 23 bp indel polymorphism. It was assumed that these polymorphisms may manipulate promoter activity leading to modified vulnerability [9].

Furthermore, three putative transcriptional factors were determined to affect transcriptional factor binding sites. These transcription factors were AP1, RP58, and SP1. SP1 is a ubiquitous protein and is occupied in the regulation of several promoters [42], and AP1 showed a role in tissues of ectodermal origin [43]. The RP58 functioned according to its purpose and was expressed generally in the brain [9,44]. It contained a POZ domain that has been expressed as an interactor with SP1. Moreover, it has been suggested that the POZ domain suppresses transcription by interfering with the DNA binding activity of SP1 [45], and transcription factor POZ protein BCL6 showed a function in B-cell demarcation and a suppression of transcription similar to RP58 [41]. In this study, reporter gene assays exposed the influence of indel polymorphisms on *PRNP* transcription sites. Remarkably, the 12 bp insertion raised the expression level in a short construct; however, the 12 bp insertion reduced the expression level in a long construct when combined with a 23 bp insertion. These results align with a model in which RP58 binds to the 5_flanking sequence of the *PRNP* gene and interacts with SP1 located within intron 1 to repress *PRNP* promoter activity. This model is consistent with other promoters that have been shown to exhibit RP58–SP1 interactions [9,41].

## 5. Conclusions

In this study, we determined the significant association between *PRNP* promoter indel polymorphisms and the BSE status in three of the investigated ruminant species. We conclude that variations in *PRNP* expression are caused by bovine promoter polymorphisms, which could lead to differences in BSE incubation time and susceptibility and which determines whether *PRNP* expression levels can be used to predict whether certain breeds may be more susceptible to BSE. We found that two indel polymorphisms in the bovine *PRNP* promoter that contain binding sites for the RP58 and SP1 transcription factors were at the expression level of *PRNP*, and data show that the *PRNP* genotype may contribute to the observed high variance in *PRNP* expression. Additionally, the findings of this study could be used to guide breed selection and breeding in order to improve the fight against BSE and the health conditions for its propagation in this region. We examined the *PRNP* expression in these breeds because of their commercial value in China and because this disease has not been diagnosed in YK, YC, and PK. These findings suggest that additional post-transcriptional studies are justified in elucidating the mechanisms underlying prion diseases.

## Figures and Tables

**Figure 1 animals-13-03627-f001:**
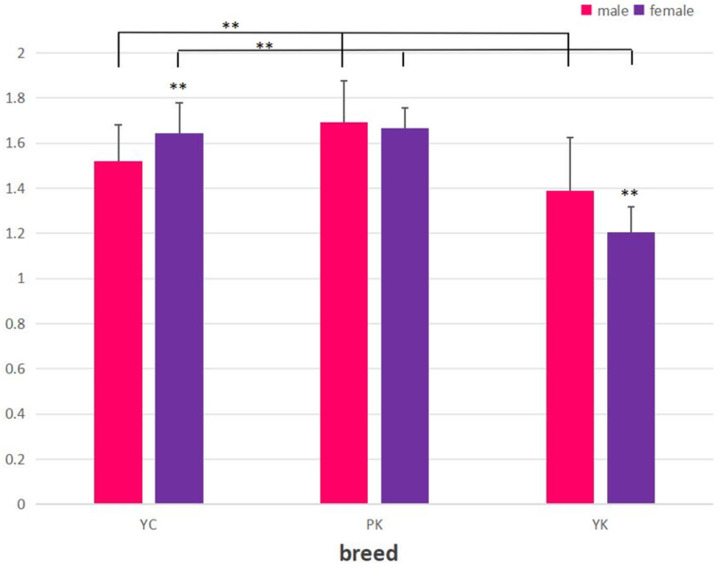
Expression of *PRNP* gene to differentiate sex. YC represents the Zhongdian Yellow cattle, PK represents the Zhongdian Yakow, and YK represents the Zhongdian Yak. ** indicates *p* < 0.01.

**Figure 2 animals-13-03627-f002:**
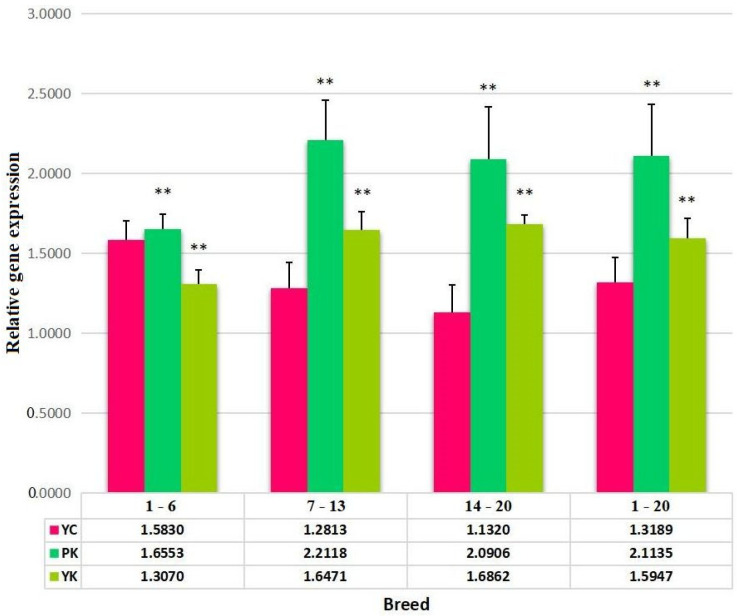
Expression of *PRNP* gene to differentiate age. YC represents the Zhongdian Yellow cattle, PK represents the Zhongdian Yakow, and YK represents the Zhongdian Yak. ** indicates *p* < 0.01. 1–6, 7–13, 14–20, and 1–20 represent the age in months.

**Figure 3 animals-13-03627-f003:**
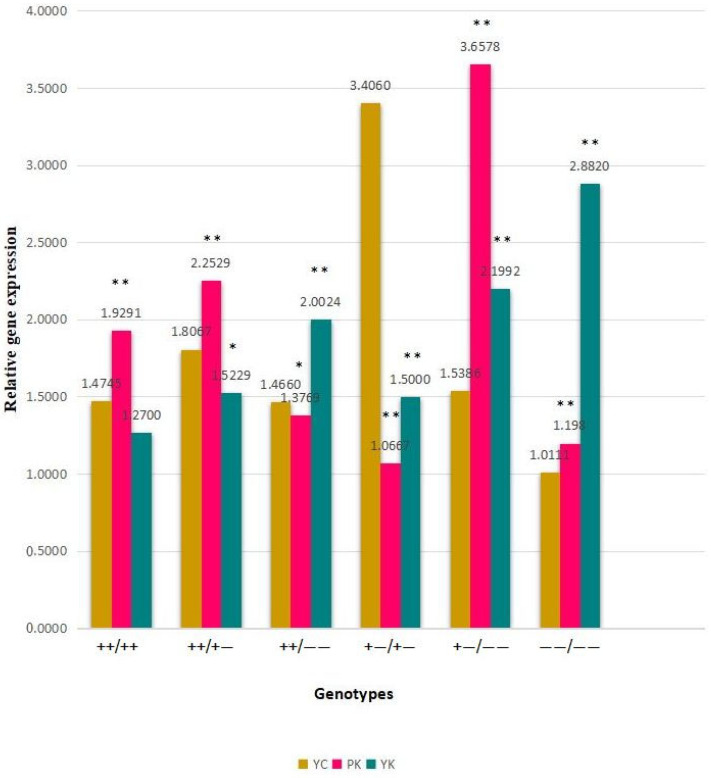
Expression of *PRNP* gene to differentiate genotype. YC represents the Zhongdian Yellow cattle, PK represents the Zhongdian Yakow, and YK represents the Zhongdian Yak. + represents insertion and − represents deletion. * indicates *p* < 0.05, ** indicates *p* < 0.01.

**Figure 4 animals-13-03627-f004:**
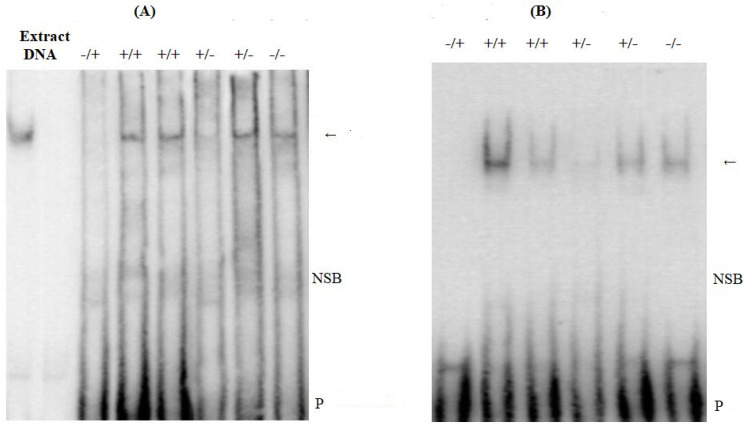
EMSA analysis: (**A**) EMSA analysis of AP1 binding to the *PRNP* promoter sequence; (**B**) EMSA analysis of SP1 binding to the *PRNP* promoter sequence. Arrow: DNA binding proteins; NSB: non-specific binding; P: free biotin-labeled probe; AP1+ Probe: 12 bp insert with AP1 domain; AP1− Probe: DNA fragment without 12 bp insert; SP1+ Probe: 23 bp insert with SP1 domain; and SP1− Probe: DNA fragment without 23 bp insert.

**Figure 5 animals-13-03627-f005:**
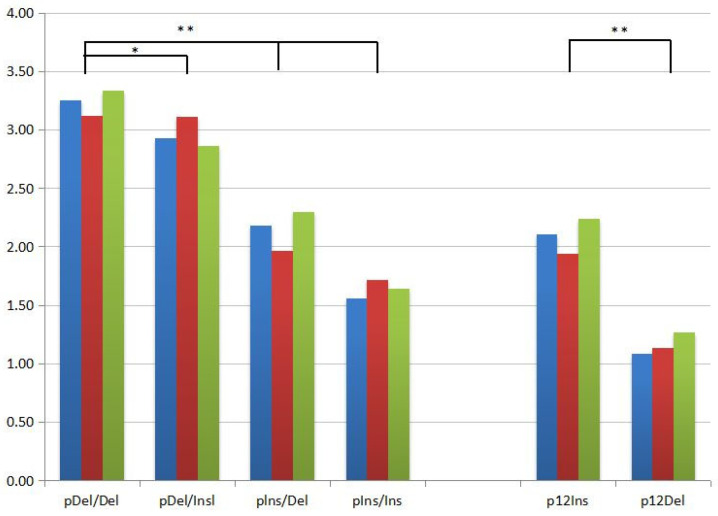
Three repeat assays for each gene reporter construction/transfection were used in this study. The positions of the Ins/Del fragments are given with the start of the luciferase reporter gene. PCR products were cloned into a pGL3-Basic Vector luciferase reporter vector (Promega) by using restriction endonucleases Mlu-I and Bgl-II for pDel/Del, pDel/Ins, pIns/Del, and pIns/Ins, or Xol-I and Bgl-II for p12Ins and p12Del. * indicates *p* < 0.05, ** indicates *p* < 0.01.

**Table 1 animals-13-03627-t001:** Genotypic and allelic frequencies of *PRNP* polymorphisms in Zhongdian Yak, Zhongdian Chinese Yakow, and Zhongdian Yellow cattle species.

Locus	Species	N	Genotypic Frequency				Allelic Frequency		
			−/−	−/+	+/+	*p*-Value	−	+	*p*-Value
23 bp indel	Zhongdian Yak	562	0.838	0.144	0.018	<0.001	0.91	0.09	<0.001
	Zhongdian Chinese Yakow	377	0.467	0.241	0.292		0.588	0.412	
	Zhongdian Yellow cattle	368	0.399	0.389	0.212		0.594	0.406	
12 bp indel	Zhongdian Yak	540	0.03	0.203	0.767	<0.001	0.131	0.869	<0.001
	Zhongdian Chinese Yakow	374	0.053	0.233	0.714		0.17	0.83	
	Zhongdian Yellow cattle	364	0.093	0.286	0.621		0.236	0.764	

**Table 2 animals-13-03627-t002:** Haplotype gene frequencies of *PRNP* polymorphisms in Zhongdian Yak, Zhongdian Chinese Yakow, and Zhongdian Yellow cattle species.

		Haplotype Frequency				
Species	N	23+/12+	23+/12−	23−/12+	23−/12−	*p*-Value
Zhongdian Yak	568	0.074	0.011	0.796	0.12	
Zhongdian Chinese Yakow	392	0.392	0.024	0.444	0.14	
Zhongdian Yellow cattle	371	0.332	0.434	0.073	0.161	<0.001

**Table 3 animals-13-03627-t003:** *PRNP* expressions on 23 bp indel for various genotypes in Zhongdian Yak, Zhongdian Chinese Yakow, and Zhongdian Yellow cattle breeds.

Species	Genotype	n	Mean *PRNP*	*p*-Value		SD	SE
				+−	−−		
Zhongdian Yak	++	3	0.5020	>0.05	>0.05	0.37074	0.21405
	+−	10	0.6058		>0.05	0.49696	0.15715
	−−	237	0.8100	>0.05		0.66493	0.04319
Zhongdian Chinese Yakow	++	44	3.8137	>0.05	>0.05	4.64625	0.70045
	+−	45	4.1594		>0.05	9.91724	1.47837
	−−	73	4.1952	>0.05		10.56636	1.23670
Zhongdian Yellow cattle	++	24	1.6849	>0.05	>0.05	1.73170	0.35348
	+−	49	2.2533		>0.05	2.54034	0.36291
	−−	92	3.6042	>0.05		10.22709	1.06625

**Table 4 animals-13-03627-t004:** *PRNP* expressions on 12 bp indel for various genotypes in Zhongdian Yak, Zhongdian Chinese Yakow, and Zhongdian Yellow cattle breeds.

Species	Genotype	n	Mean *PRNP*	*p*-Value		SD	SE
				+−	−−		
Zhongdian Yak	++	225	0.7818	>0.05	<0.05	0.65246	0.04350
	+−	15	0.8171		>0.05	0.62599	0.16163
	−−	10	1.1373	>0.05		0.77794	0.24600
Zhongdian Chinese Yakow	++	126	3.6845	>0.05	>0.05	6.84282	0.60961
	+−	30	6.1033		>0.05	15.83813	2.89163
	−−	6	2.3135	>0.05		1.41004	0.57565
Zhongdian Yellow cattle	++	125	3.2614	>0.05	>0.05	8.89117	0.79525
	+−	29	2.1357		<0.05	2.04878	0.38045
	−−	11	1.1654	<0.05		1.08766	0.32794

**Table 5 animals-13-03627-t005:** *PRNP* expression for various diplotypes in Zhongdian Yak, Zhongdian Chinese Yakow, and Zhongdian Yellow cattle species.

Species	Diplotype	N	Mean *PRNP*	SD	SE	*p*-Value
Zhongdian Yak	++/++	3	1.2700	0.9382	0.54169	>0.05
	++/+−	7	1.5229	1.4407	0.54453	
	++/−−	215	2.0024	1.6674	0.11372	
	+−/+−	3	1.5000	1.0516	0.60715	
	+−/−−	12	2.1992	1.6931	0.48877	
	−−/−−	10	2.8820	1.9694	0.62277	
Zhongdian Chinese Yakow	++/++	43	1.9291	2.3218	0.35408	>0.05
	++/+−	38	2.2529	5.3371	0.86579	
	++/−−	45	1.3769	1.8228	0.27172	
	+−/+−	6	1.0667	0.9470	0.38662	
	+−/−−	23	3.6578	8.9185	1.85964	
	−−/−−	5	1.1980	0.7694	0.34408	
Zhongdian Yellow cattle	++/++	22	1.4745	1.5404	0.32841	>0.05
	++/+−	42	1.8067	2.1233	0.32763	
	++/−−	57	1.4660	1.3140	0.17404	
	+−/++	2	1.1500	0.9192	0.65000	
	+−/+−	5	3.4060	2.7244	1.21839	
	+−/−−	22	1.5386	1.3966	0.29776	
	−−/+−	2	0.9450	0.5586	0.39500	
	−−/−−	9	1.0111	1.0248	0.34159	

## Data Availability

The data presented in this study are available on request from the corresponding author.

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
