# Peer review of "Comparative Analysis of PRNP Gene Indel Polymorphism and Expression among Zhongdian Yellow Cattle, Zhongdian Yak, and Their Hybrids"

_animals, 2023, doi:10.3390/ani13233627_

Round 1

Reviewer 1 Report

Comments and Suggestions for Authors

This article reveals PRNP gene indices polymorphism and 2 expression among Chinese Yellow Cattle, but there are still many defects in the articlesuggesting author must perform substantive modifications.

 1. How was the samples of medulla oblongata tissue collected? Is it in compliance with animal ethics confirm? Need to supplement ethical statement.

 2. It is recommended to change the "Yellow cattle" in the title to "Zhongdian Yellow cattle".

3. The results of the main text and tables are completely repetitive, such as Table 4 and sections 230-240 of the main text, etc. Suggest concentrating one of them.

 4. There are multiple PRNP primers in Table 2. Please indicate the respective uses of the primers in the list. Otherwise, these primers are chaotic.

 5. How is haplotype constructed? Supplement in the method section.

 6. Line287-296, Table 7: The indel of 12bp in intron1 is cut during post-transcriptional processing, how can qPCR detection be performed using cDNA as a template? This is very absurd.

7. No significant difference test was conducted between the groups, such as the results in Figures 1-3 and Tables 6-8. All results involving gene expression should have differential expression analysis results.

 8. What do these symbols in Table 9 represent? Suggest using pictures to represent.

9. The conclusion needs to be further condensed and the significance of the research results should be explained.

Author Response

Reviewers 1:

Comment 1: How was the samples of medulla oblongata tissue collected? Is it in compliance with animal ethics confirm? Need to supplement ethical statement.

Response 1: I really appreciate your comment. We obtained medulla oblongata tissue samples from slaughterhouse animals (slaughter animals), including 562 YK, 368 YC and 377 PK in Shangri-La City, China.

Additionally we also already described the ethical statement in the manuscript in the last after funding information (Institutional Review Board Statement: All the animal manipulations were approved by the Ethics Committee of Experimental Animal of Yunnan Agricultural University (Agreement No. AC.2019-2021)).

Comment 2: It is recommended to change the "Yellow cattle" in the title to "Zhongdian Yellow cattle".

Response 2: Thank you for the comments. I have modified in the manuscript as marked yellow according to your comments.

Comment 3: The results of the main text and tables are completely repetitive, such as Table 4 and sections 230-240 of the main text, etc. Suggest concentrating one of them.

Response 3: Thank you very much for your comment. I have modified in the manuscript accordingly.

Comment 4: There are multiple PRNP primers in Table 2. Please indicate the respective uses of the primers in the list. Otherwise, these primers are chaotic.

Response 4: Thank you so much for your comment. In this study, we organized the research plan with 2 technical routes. I also added the additional supplementary tiff file where showed the technical route of my study.

In this study, in one side we done PCR and investigated the indel polymorphism (Genotyping with respect to the 23-bp indel within the promoter region and 12-bp indel within the first intron were amplified by polymerase chain reaction (PCR) using primer pairs in Table 2). Additionally I have replaced Table 1,2 ,3 in supplementary tables (Table S1, Table S2, Table S3).

And other side measured the mRNA expression of PRNP gene by real-time fluorescence quantitative PCR (Real-time PCR standard curves for each of the PRNP and an endogenous control actin beta gene (ACTB) were constructed by PCR products including the target sequence and its flanking sequence as a template. Gene expression levels of the bovine PRNP and ACTB were quantitatively measured by the real-time PCR using primer pairs also listed in Table S2)

Comment 5: How is haplotype constructed? Supplement in the method section.

Response 5: Thank you for your comment. I have added information in the method section (Statistical analysis) and marked as a yellow in the manuscript.

Comment 6: Line287-296, Table 7: The indel of 12bp in intron1 is cut during post-transcriptional processing, how can qPCR detection be performed using cDNA as a template? This is very absurd.

Response 6: Thank you for comment. Here in Table 7, In order to evaluate PRNP expression among the genotypes of the 12-bp locus, we selected animals with diplotypes −−/++, −−/+− and −−/−− .

Diplotypes for the two loci were created using the genotypes. In the analysis of gene expression, animals possessing the chosen diplotypes were used. The diplotypes were chosen taking into consideration the sample sizes and the goals of our research. Sampling of the medulla oblongata tissue was limited to 30–40 samples per sampling day in order to prevent analysis of animals of the same origin (family relations, sex, age, or farms).

Comment 7: No significant difference test was conducted between the groups, such as the results in Figures 1-3 and Tables 6-8. All results involving gene expression should have differential expression analysis results.

Response 7: Thanks for comment. I have done significant difference tests and added P>value in the tables and figures. Here also I replaced tables 4-8 into 1-5.

Comment 8: What do these symbols in Table 9 represent? Suggest using pictures to represent.

Response 8: I greatly appreciate your comments. We reported in the manuscript that all alleles for each polymorphic location were shown for transcription factor-binding sites, which led to the identification of 44 divergences in candidate transcription factor-binding sites. In this study we used three transcription factor-binding sites.

Below table 9 written note where just showed (Note:The representative AP-1, RP58 and SP-1 transcriptional factors are indicated with red arrows).

According to Table 9, I already modified this table in manuscript as supplementary Table S3.

Comment 9: The conclusion needs to be further condensed and the significance of the research results should be explained.

Response 9: Thank you so much for your comment. I have modified accordingly.

All the points raised by the reviewers for the revision of the manuscript have been duly considered and could be found in the coloured text in the main file of the manuscript. new updated Figures and tables I've positioned it after the references at the end of the manuscript.

Reviewer 2 Report

Comments and Suggestions for Authors

This study provides valuable insights into the role of PRNP gene polymorphisms in the expression of bovine prion protein and their potential contribution to BSE susceptibility. The authors have clearly defined the objectives and methodology, and the results are well presented with appropriate statistical analysis. However, there are a few areas that could be improved upon in future revisions.

There is a confusion in the layout of the article: the subscript Breed in Fig.2 is not centered. The superscripts of images A and B in Fig.4 are inconsistent. Some chaotic red arrows appear in Table 9. Pay attention to the layout of the table and try to be on the same page to ensure cleanliness and aesthetics.

Comments on the Quality of English Language

I suggest that you thoroughly revise the English language usage throughout the manuscript. There are several instances of awkward phrasing, incorrect word choice, and grammatical errors that need to be addressed. It is important to ensure that the language used is clear, concise, and easily understandable by a broad audience. Also, I recommend that you provide more context and background information in certain sections of the manuscript. This will help readers better understand the significance of your findings and the relevance of your research to the field. Additionally, it may be helpful to include more specific examples and case studies to support your arguments.

Author Response

Reviewers 2:

Comment 1: There is a confusion in the layout of the article: the subscript Breed in Fig.2 is not centered. The superscripts of images A and B in Fig.4 are inconsistent. Some chaotic red arrows appear in Table 9. Pay attention to the layout of the table and try to be on the same page to ensure cleanliness and aesthetics

Response 1: I really appreciate your comments. This is the article's initial layout. After reviewing it, we will review it and organize it properly.

According to Table 9, I already modified this table in manuscript as supplementary Table S3.

Comment 2: I suggest that you thoroughly revise the English language usage throughout the manuscript. There are several instances of awkward phrasing, incorrect word choice, and grammatical errors that need to be addressed. It is important to ensure that the language used is clear, concise, and easily understandable by a broad audience. Also, I recommend that you provide more context and background information in certain sections of the manuscript. This will help readers better understand the significance of your findings and the relevance of your research to the field. Additionally, it may be helpful to include more specific examples and case studies to support your arguments.

Response 2: Thank you so much for your comments and recommendation. I have revised English language and awkward phrasing and grammatical errors marked as red in the manuscript. We have already described some background information in the beginning of the discussion section so that readers can easily understand the significance of the study, in accordance with the additional context provided in the background section. So, please let me know if more context is needed, and I will provide it.

Reviewer 3 Report

Comments and Suggestions for Authors

In this study, the authors have examined the genotype, allele, haplotype, mRNA expression, and Transcription factors binding of bovine PRNP gene in 562 Zhongdial Yak, 368 Zhongdian Yellow cattle, and 377 Zhongdian Yakow. The authors suggest that PRNP genotype contributes to high variation of PRNP expression.

The data are clearly presented and the interpretation is reasonable.

There are some minor points that the authors may consider:

1) In this manuscript, there are too many tables and figures. Table 1 can be deleted and described in Materials and methods.

2) In Table 4, please provide a p-value for genotype and allele frequencies among each species.  

3) In Table 5, please provide a p-value for haplotype frequencies among each species.

4) Tables 4 and 5 can be combined.

5) Figures 1, 2 and 3 can be combined into one figure.

6) In Figures 1, 2, and 3, please provide a p-value between each group.

7) In Figures 1, 2, and 3, the figure legend should be described in more detail, including the number of samples used or which organ or cell did you use ?

8) Tables 6 and 7 can be combined.

9) In Tables 6 and 7, please provide a p-value between each group.

10) In Table 8, please provide a p-value between each group.

11) Figures 4A and 4B can be combined into Figure 4.

12) In Figures 4A and 4B, the figure legend should be described in more detail.

13) In Figure 5, please provide a p-value between each group.

14) In Figure 5, the figure legend should be described in more detail.

15) For the three genotypes, 12 bp and 23 bp, the experimental data should be shown as supplementary figures. Ex) The electropherograms of DNA Sequencing or Gel electrophoresis photo of PCR products,, etc.

Author Response

Reviewers 3:

Comment 1: In this manuscript, there are too many tables and figures. Table 1 can be deleted and described in Materials and methods.

Response 1: I greatly appreciate your comment. I have updated Table S1 in the manuscript and replaced Table 1 in Supplementary Table S1.

Comment 2: In Table 4, please provide a p-value for genotype and allele frequencies among each species.

Response 2: I really appreciate your comment. I have included the P-value and modified in the manuscript accordingly. Table 4 replaced into Table 1.

Comment 3: In Table 5, please provide a p-value for haplotype frequencies among each species.

Response 3: Thank you for your comment. I have included the P-value and modified in the manuscript accordingly. Table 5 replaced into Table 2.

Comment 4: Tables 4 and 5 can be combined.

Response 4: Thank you very much for your comment. I greatly appreciate your feedback. Actually, Tables 1, 2, and 3 have been replaced with supplemental tables. Thus, I did not merge Tables 4 and 5. Each table in this manuscript needs to be explained separately.

Comment 5: Figures 1, 2 and 3 can be combined into one figure.

Response 5: I really appreciate your comment. I think it's preferable to keep each figure apart. Same as here also each figure in this manuscript needs to be explained separately.

Comment 6: In Figures 1, 2, and 3, please provide a p-value between each group.

Response 6: Thank you for your comment. I have included the P-value and modified in the manuscript accordingly.

Comment 7: In Figures 1, 2, and 3, the figure legend should be described in more detail, including the number of samples used or which organ or cell did you use ?

Response 7: Thank you very much for your comment. I have described figure legends in more detail in manuscript accordingly.

Comment 8: Tables 6 and 7 can be combined.

Response 8: I did not combine. Tables 6 and 7. This manuscript needs to have a separate explanation for each table.

.

.Comment 9: In Tables 6 and 7, please provide a p-value between each group.

Response 9: Thank you for your comment. I have included the P-value and modified in the manuscript accordingly. Table 6 and 7 replaced into Table 3 and 4.

Comment 10:  In Table 8, please provide a p-value between each group.

Response 10: Thank you for your comment. I have included the P-value and modified in the manuscript accordingly. Table 8 replaced into Table 5.

Comment 11: Figures 4A and 4B can be combined into Figure 4.

Response 11: Thanks you for you comment. Here I combined Figures 4A and 4B into Figure 4.

Comment 12: In Figures 4A and 4B, the figure legend should be described in more detail.

Response 12: I have described figure legends in more detail in manuscript accordingly.

Comment 13:  In Figure 5, please provide a p-value between each group.

Response 13: Thank you for your comment. I have included the P-value and modified in the manuscript accordingly.

Comment 14: In Figure 5, the figure legend should be described in more detail.

Response 14: I have described figure legends in more detail in manuscript accordingly.

All the points raised by the reviewers for the revision of the manuscript have been duly considered and could be found in the coloured text in the main file of the manuscript. new updated Figures and tables I've positioned it after the references at the end of the manuscript.

Round 2

Reviewer 1 Report

Comments and Suggestions for Authors

Authors have made in-depth revisions to the manuscript according to the comments of the reviewers. I suggest that the manuscript in the curent version could be acceptable for publication.